# ONLINE RUBRICS ELICITATION FROM PAIRWISE COMPARISONS

## ABSTRACT

Rubrics provide a flexible way to train LLMs on open-ended long-form answers where verifiable rewards are not applicable and human preferences provide coarse signals. Prior work shows that reinforcement learning with rubric-based rewards leads to consistent gains in LLM post-training. Most existing approaches rely on rubrics that remain static over the course of training. Such static rubrics, however, are vulnerable to reward-hacking type behaviors and fail to capture emergent desiderata that arise during training. We introduce Online Rubrics Elicitation (OnlineRubrics), a method that dynamically curates evaluation criteria in an *online* manner through pairwise comparisons of responses from current and reference policies. This online process enables continuous identification and mitigation of errors as training proceeds. Empirically, this approach yields consistent improvements of up to 8% over training exclusively with static rubrics across AlpacaEval, GPQA, ArenaHard as well as the validation sets of expert questions and rubrics. We qualitatively analyze the elicited criteria and identify prominent themes such as transparency, practicality, organization, and reasoning.

## 1 INTRODUCTION

Recent advances in reinforcement learning are reshaping the traditional post-training recipe. The work of Guo et al. (2025) demonstrated that supervised fine-tuning on instructions can be skipped altogether, with policies (e.g. R1-Zero) trained directly via reinforcement learning, disrupting the way researchers think about post-training. Since then, much of the focus has shifted towards reinforcement learning. However, R1-Zero was trained only using verifiable rewards; the final response is easily gradable, think of a number or code snippet with unit tests, which is only applicable to limited domains.

To accommodate broader settings, rubric-based scoring for reinforcement learning emerges as an alternative way for reward modeling, particularly for long-form responses (Viswanathan et al., 2025; Gunjal et al., 2025; Huang et al., 2025; Anugraha et al., 2025). Rubrics are comprised of a list of input-specific criteria that characterizes an ideal response; one example criterion in the finance domain is *"States shocking basis causes nonlinear effects in margin calls"*. Each criterion has an importance weight: satisfying positively weighted criteria yields reward, while satisfying negatively weighted criteria yields penalty. During training, an LLM-based grader evaluates a response against each criterion in the rubric, producing binary satisfaction scores; and the overall score is the weighted average of these grades. This framework extends reinforcement learning to both verifiable and non-verifiable aspects of responses, spanning generalist and expert domains alike.

Rubrics often emphasize the desired behaviors with less coverage of undesired properties. *Offline rubrics* created apriori, human-written or synthetic, cannot realistically cover every unexpected (and desired) pattern. Fixed checklists (Wang et al., 2024) to enforce generally helpful patterns e.g. truthfulness, instruction following or relevance, fall short in preventing nuanced errors. For example, Huang et al. (2025) identifies "self-praising" as one emerging pattern during reinforcement learning from rubrics, think of including *"The following advice is the most relevant"* as part of the response; these praises often fool the LLM-based verifier into believing that the given response is indeed relevant. Such patterns are especially difficult for generic "catch-all" rubrics to reveal when they are sample-specific. Moreover, correct traits in some generations can go unnoticed if not readily rewarded by the existing offline rubrics.

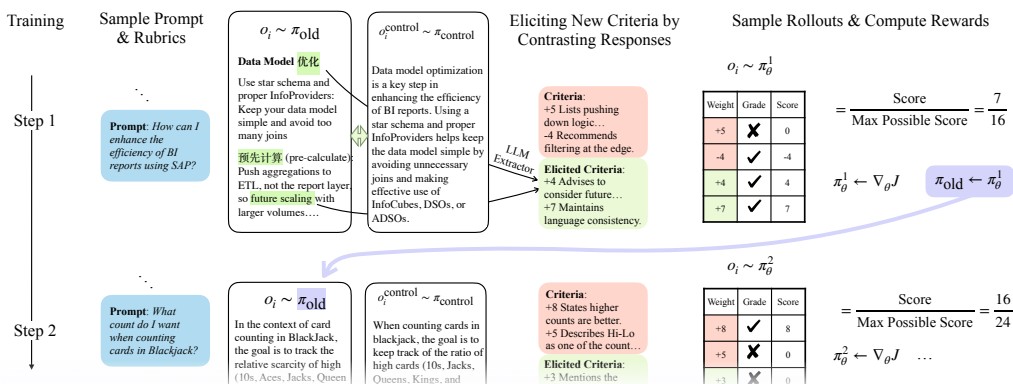

Figure 1: At any step during training, OnlineRubrics starts off by considering a pair of responses, one of which is from the current policy before updates and another from a *control* model e.g. reference model. We follow with LLM-based rubrics elicitation and deduplication steps to generate a set of criteria. These criteria along with existing criteria (e.g. human-written or synthetic) are used to create the reward in the policy gradient algorithm.

We introduce OnlineRubrics, a framework for eliciting evaluation criteria dynamically via pairwise comparisons. OnlineRubrics leverages a pair of responses in creating additional criteria where the responses are sampled from the current policy and a control model. Our work, as depicted in Fig. 1, is inspired by the large body of literature on preference learning (Akrour et al., 2011; Fürnkranz et al., 2012; Schoenauer et al., 2014) and pairwise reward modeling (Christiano et al., 2017; Stiennon et al., 2020; Ouyang et al., 2022). While LLMs are imperfect judges of quality (Gu et al., 2024), we found that pairwise comparisons are easier to make for the models when identifying new criteria than directly making a quality assessment or creating new criteria by considering a single response (point-wise elicitation). The additional criteria simply augments the existing rubric, enabling seamless integration of OnlineRubrics with any rubric-based scoring mechanism.

In training and evaluating our approach, we curate two datasets for expert (scientific use-cases) and generalist domains. We additionally conduct out-of-distribution evaluations using public benchmarks, comparing different approaches to reward estimation. OnlineRubrics results in absolute gains of up to 25% over the initial instruct model across various benchmarks including GPQA-Diamond, GSM8K, AlpacaEval, and Arena-Hard.

## 2 RELATED WORK

**Reward Modeling** The dominant paradigm in LLM aligment is to learn a reward function from feedback. Foundational work in Reinforcement Learning from Human Feedback (RLHF) established the use of pairwise preference comparisons–preferred over less robust pointwise scores–to train an explicit reward model (Ouyang et al., 2022; Stiennon et al., 2020). This process was later simplified by methods like Direct Preference Optimization (DPO; Rafailov et al. (2023)), which bypasses the explicit reward model and optimizes policies directly on preference data. Methods for generating feedback have also advanced: Bai et al. (2022), for example, pioneered the use of AI feedback (RLAIF) by leveraging a fixed set of principles for model self-feedback. More recently, research has focused on improving the reward model's intrinsic capability. Liu et al. (2025) established inference-time scaling laws for generalist reward models, boosting performance with added computation, while Whitehouse et al. (2025) incentivizes faithful evaluation by training LLM judges to generate reasoning.

**Multi-Objective Alignment** Beyond single-reward formulations, recent research has explored *multi-objective RLHF* approaches that optimize across several criteria simultaneously. Safe RLHF (Dai et al., 2023) decouples helpfulness and harmlessness rewards and balances them using con-

strained optimization. Gradient-Adaptive Policy Optimization (GAPO) (Li et al., 2025) employs multiple-gradient descent to achieve Pareto-optimal trade-offs across competing objectives, while Lu et al. (2025) proposes dynamically adjusting reward weights online. Similarly, conditional reward modeling (Cai et al., 2024) allows a single reward model to flexibly apply different principles depending on context in training their evaluator LLM. These works highlight growing recognition that LLM alignment requires balancing diverse objectives which is closely related to our focus on dynamically eliciting new rubrics.

**Verifiable Rewards** While preference-based rewards provide flexible but often fuzzy signals, verifiable rewards offer exact supervision whenever the outcome can be automatically checked. Reinforcement Learning with Verifiable Rewards (RLVR) improves reasoning by optimizing policies against automatically checkable outcomes, such as numeric answers or unit-tested code. Recent work has shown its effectiveness across various domains: DeepSeek-R1 (Guo et al., 2025) and General-Reasoner (Ma et al., 2025) achieved strong results on benchmarks such as GSM8K (Cobbe et al., 2021), MMLU (Hendrycks et al., 2021), and GPQA (Rein et al., 2024). In medicine, Zhang et al. (2025) enabled a 3B model to reach expert-level performance. Foundational studies confirm that RLVR incentivizes correct reasoning processes, not just correct answers Wen et al. (2025). Despite these strengths, RLVR does not extend to open-ended domains where correctness cannot be automatically verified.

**Evaluating and Training with Rubrics** Recent work has extended the concept of verifiable rewards from domains like math and coding to more open-ended tasks by using rubrics for structured evaluation. This rubric-based approach has been adopted in various benchmarks for both expert (Arora et al., 2025; Starace et al., 2025) and generalist domains (Deshpande et al., 2025). Beyond evaluation, rubrics are now increasingly used as direct reward signals for reinforcement learning. Using structured rubrics as a direct reward has proven effective in both expert reasoning (Gunjal et al., 2025) and generalist alignment (Viswanathan et al., 2025). A diverse set of rubrics has also been used to train a single, robust reward model that generalizes across various domains (Anugraha et al., 2025). Our work complements these methods; instead of using a static rubric or training a rubric-agnostic model, OnlineRubrics dynamically augments criteria online to adapt to the policy's emergent behaviors.

## 3 BACKGROUND

Rubrics are often used as drop-in replacement for rewards in any policy gradient learning algorithm.

### 3.1 TRAINING SETUP

In this work, we used GRPO algorithm (Shao et al., 2024) which maximizes the following objective

$$\mathcal{L}_{\text{GRPO}}(\theta) = \mathbb{E}_{i \sim \mathcal{D}, j \sim \mathcal{G}_i} \left[ \min \left( r_{i,j}(\theta) \, \hat{A}_{i,j}^{\text{group}}, \, \text{clip}\left( r_{i,j}(\theta), \, 1-\epsilon, \, 1+\epsilon \right) \hat{A}_{i,j}^{\text{group}} \right) - \beta \mathbb{D}_{KL} \left( \pi_\theta || \pi_{ref} \right) \right] \quad (1)$$

where $r_{i,j}(\theta) = \frac{\pi_\theta(o_{i,j}|x_i)}{\pi_{\theta_{\text{old}}}(o_{i,j}|x_i)}$ is the probability ratio, and advantages are calculated as normalized rewards:

$$\hat{A}_{i,j}^{\text{group}} = \frac{R_j - \text{mean}(\mathbf{R})}{\text{std}(\mathbf{R})} \quad (2)$$

where $\mathcal{D} = \{x_i, \mathcal{C}_i\}$ is the set of training prompts and criteria, $j$ indexes the output samples $o_j$ from the group $o_j \sim \mathcal{G}_i$, $\pi_{\theta_{\text{old}}}$ is the policy before the update, $\pi_\theta$ the target policy. The rewards are computed independently for each $o_j$ in the group and denoted by $\mathbf{R} = \{R_1, R_2, \ldots, R_G\}$ where $G$ is the group size. In this work, we will assume that the true reward $U$ can be modeled as a function of latent criteria and argue in Section 4.2 that for optimal modeling of the true reward all criteria should be elicited.

---

**Algorithm 1:** Online Rubric Eliciting (OnlineRubrics)

---

**Input:** Policy $\pi_\theta$, control policy $\pi_{\text{control}}$, dataset $\mathcal{D}$, extraction prompt $P_e$, hyperparameter $M$

**for** $step = 1, 2, \ldots, N$ **do**

    Sample prompts and criteria $\{x_i, \mathcal{C}_i\}$ from $\mathcal{D}$;

    Update $\pi_{\text{old}} \leftarrow \pi_\theta$;

    Generate $M$ candidate responses $\{o_{i,j}\}$ using $\pi_{\text{old}}$;

    Generate $M$ candidate responses $\{o_{i,j}^{\text{control}}\}$ using $\pi_{\text{control}}$;

    Initialize $C_i^e \leftarrow \emptyset$;

    **for** $k = 1, 2, \ldots, M$ **do**

        Extract new criteria $C_{i,k}^e \sim \text{LLM}_{\text{extract}}(x_i, o_{i,k}, o_{i,k}^{\text{control}}; P_e)$;

        $C_i^e \leftarrow C_i^e \cup C_{i,k}^e$;

    De-duplicate $C_i^e$;

    Compute rewards using Eq. (3) and $\mathcal{C} = \mathcal{C}_i \bigcup C_i^e$;

    Compute group advantages $\hat{A}_{i,j}$ Eq. (2);

    Update $\theta$ via policy gradient by maximizing Eq. (1)

---

## 3.2 Rubric Based Rewards

In RLHF, reward signals in LLM training are traditionally modeled after human preferences with an explicit reward model in PPO (Schulman et al., 2017) and GRPO or implicitly in DPO. In the case of queries where quick verification of the final answer is possible (i.e. numeric or short answer), exact match replaces human preferences for reward. More recently, rubrics for evaluating long-form answers are being used for calculating final scores (Gunjal et al., 2025; Huang et al., 2025; Viswanathan et al., 2025) where an LLM-based grader (denoted by $\text{LLM}_{\text{grader}}$) evaluates a response against each criteria to compute $R_j$ in Eq. (3):

$$R_j = q\Big(\text{LLM}_{\text{grader}}\Big(o_j, x_i, \mathcal{C}_i\Big)\Big) \tag{3}$$

where $\mathcal{C}_i = \{(c_1, w_1), (c_2, w_2), \ldots, (c_d, w_d)\}$ is a collection of criteria with corresponding importance weights that describe an ideal response to the prompt, and $q$ is an aggregation function. The judge $\text{LLM}_{\text{grader}}$ (Zheng et al., 2023) evaluates the output $o_j$ against each criterion in $C_i$ and produces a list of binary outcomes which are then reduced to a single scalar value by $q$ using the weights, if applicable. In this work we implement the reduction function as a weighted sum of the grades normalized by the total possible score:

$$q(x, o, \mathcal{C}) = \frac{w^\top \text{LLM}_{\text{grader}}(x, o, \mathcal{C})}{\sum_{k:w_k > 0} w_k} \tag{4}$$

where $\text{LLM}_{\text{grader}}(x, o, \mathcal{C}) \in \{0, 1\}^d$ is the binary grades corresponding to each criterion.

## 4 Online Rubric Elicitation

Rubric-based reward calculation provides richer feedback than reward-model-based post-training, yet it fails to mitigate the problems that might emerge during policy gradient updates. Specifically, we observe that initial rubrics tend to represent the desired qualities of an ideal response while putting less emphasis on describing undesired qualities. For example, when the prompt is *How can I travel to San Francisco from San Jose?* and the rubric is *(+9, The response mentions Caltrain)* both responses *Caltrain is good, on the way back consider renting a car* and *Caltrain and renting a car are both good options* get the full score while the former has redundancy about the return options. Such mishaps may only be detected as they arise during rollouts. Moreover, emerging qualities that are not currently rewarded by the existing rubric set will be overlooked by the algorithm.

We propose a novel method called OnlineRubrics that leverages pairwise comparison of candidate responses to derive novel criteria–OnlineRubrics is designed to capture potential errors and identify

```
You are given a prompt and pair of responses to the same prompt. Your task is to identify their
differences not already covered by the existing rubrics. [truncated] First, analyze both responses
to identify the differences. Then, transform these observations into new evaluation criteria if
they're not already covered by existing rubrics. This is very important: any rubric that you
introduce should be based on one of the responses. Do not use your own knowledge to introduce new
criteria that are not based on one of the responses.

Focus on criteria that distinguish genuinely helpful responses from those gaming the system.
[truncated] Assign a positive weight (integer) to each of the new criteria based on the relative
importance of the criterion to the existing criteria.
If no meaningful new criteria are needed, return an empty list.

{{Existing Rubric}}
{{Response A}}
{{Response B}}
```

Figure 2: Abbreviated system prompt template used for $\text{LLM}_{\text{extractor}}$, see full prompt in Fig. 8.

useful features. The approach simply augments the set of offline criteria i.e. the portion of the rubric that is created apriori for the specific prompt, with more criteria derived during the training. Our approach is different from recent work that uses a fixed set of criteria (or checklists) (Anugraha et al., 2025) for multiple data points or other procedures to extract rubrics in a pointwise manner by simply considering a prompt (Huang et al., 2025). OnlineRubrics drives insights from the pairwise reward modeling literature (Bradley & Terry, 1952; Stiennon et al., 2020; Ouyang et al., 2022).

### 4.1 LLM-BASED CRITERIA ELICITATION

OnlineRubrics begins with an initial set of *offline criteria* $\mathcal{C}_i$ that may be provided by human annotators or created synthetically. During policy training, at step $t$ before any updates, given a prompt $x_i$ we sample a set of candidate responses from an *control* policy (e.g. the initial policy, $\pi_{\text{ref}}$, or the policy from the previous step $\pi_{\text{old}}$) and the current policy $\pi_\theta^t$. We define an LLM-based rubric extractor $\text{LLM}_{\text{extractor}}$ conditioned on the system prompt $P_e$ (see Figure 2) whose task is to identify the differences between a pair of responses $(o_{i,j}, o_{i,j}^{\text{control}})$ sampled from the current and control policies, respectively, and turn them into useful criteria and corresponding weights. $\text{LLM}_{\text{extractor}}$ is instructed to provide references within the responses to where the difference appears. We repeat this procedure independently for each prompt in the batch and augment their corresponding rubrics with the new criteria before the policy parameter update. We provide the procedure in Algorithm 1.

We adopt a two-step approach for criteria elicitation; in the first step, we ask $\text{LLM}_{\text{extractor}}$ to enumerate the meaningful differences between a pair of responses with references to where these differences arise in the responses. In the second stage, we reduce the criteria that are duplicates or overlap significantly to avoid redundancy following our desiderata Section 5. The system prompt template used to extract rubrics is given in Figure 2 and the deduplication prompt is available in Figure 9. By default, we compare eight pairs of rollouts from each of the control and current policies and extract about eight criteria at the end of the procedure.

**OnlineRubrics Variants** We experiment with two variants for where we change the source of alternative responses $\pi_{\text{control}}$ with one of $\pi_{\text{ref}}$ or $\pi_{\text{old}}$. We hypothesize that setting sampling the responses from the $\pi_{\text{old}}$ makes the reward-hacking behaviors less likely to surface, since both candidate and control rollouts are sampled from the same distribution, hence subject to the same failure modes. That said, we empirically observe in Table 2 that this version also performs quite strongly compared to the setting $\pi_{\text{control}} = \pi_{\text{ref}}$.

### 4.2 A FORMAL MOTIVATION FOR ONLINERUBRICS

Let $f$ be the grades from $\text{LLM}_{\text{grader}}$ for the prompt, response and criteria triplet $(x, o, \mathcal{C})$ such that $f(x, o, \mathcal{C}) \in \{0, 1\}^d$ where $\mathcal{C}$ and $w$ are the set of criteria and weights and $d$ is the size of the criteria. Let also $\mathcal{C}^E$ (explicit) and $\mathcal{C}^I$ (implicit) to denote to the set of criteria in the rubric and those not in the rubric, respectively, and $f_E(x, o)$ to indicate the binary grades for the output $o$ under criteria $\mathcal{C}^E$.

**Proposition 1.** *Suppose that*

- $\mathcal{C}^*$ *is the set of true criteria.* $f_*$ *can be split into* $f_* = (f_E, f_I)$ *and* $\mathcal{C}^* = (\mathcal{C}^E, \mathcal{C}^I)$.

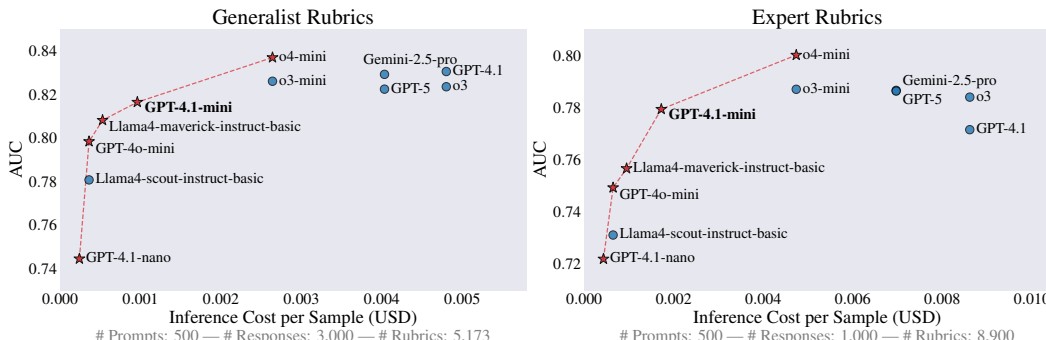

Figure 3: Performance of different LLM graders. AUC score is calculated using the receiver operating characteristic (ROC) curve. The best grader is the one with the highest AUC score and the lowest inference cost per sample. Models on the Pareto frontier (shown as a red dotted line) are the best trade-off between the two metrics. We choose GPT-4.1-mini as our default grader, balancing alignment quality with inference cost.

- *The true reward is $U(x, o) = w_E^\top f_E(x, o, \mathcal{C}^E) + w_I^\top f_I(x, o, \mathcal{C}^I)$ and the estimated reward $R_t(x, o) = w_E^\top f_E(x, o)$ at step $t$.*
- *Assuming GRPO style updates, the gradient under the true reward then would be $g_U = \mathbb{E}[\nabla_\theta \log \pi_\theta(o|x) U(x, o)]$ and the estimated gradient $g_{R_t} = \mathbb{E}[\nabla_\theta \log \pi_\theta(o|x) R_t(x, o)]$*

*Then,*

$$\|g_U - g_{R_t}\|_2 \leq \sqrt{\mathbb{E}\left[\left\|\nabla_\theta \log_{\pi_\theta}\right\|^2\right]} \|w_I\|_1$$

Proposition 1 shows that the difference between the gradient steps is upper-bounded by $\|w_I\|_1$ times the expected squared norm of the policy score function. Hence, augmenting the rubric to better approximate the true criterion set reduces the variance term, leading to an improved stability and sample efficiency during training. That said, OnlineRubrics should be viewed as a step toward tightening the upper bound on the implicit, unmodeled mass $\|w_I\|_1$, rather than a complete recovery of the true criteria set. Proof is given in Appendix A.

## 5 DATASETS

We trained OnlineRubrics with two collected rubric datasets: Generalist Rubrics and Expert Rubrics. Generalist Rubrics consists of real-world, single-turn prompts contributed with consent by users and curated to be safe, rubric-eligible, and generalist in scope. For each prompt, human annotators authored a prompt-specific rubric composed of weighted, binary-checkable criteria.

Expert Rubrics extends the same rubric framework to expert-authored problem sets across Physics, Chemistry, Biology, and Math. Each task bundles a prompt, an expert grading rubric with binary-evaluable and weighted criteria, sample model responses, and detailed rubric ratings. We use a subset of both datasets as evaluation sets and exclude from training. Table 1 shows the statistics of the datasets. On average, Generalist set contains 10.4 rubrics per sample and Expert set contains 18.0 rubrics per sample. Across both datasets, rubrics are human-written and follow the same annotation principles: criteria are *Mutually Exclusive & Collectively Exhaustive, Atomic, Objective, and Self-Contained*; ensuring they can be verified reliably and used as dense reward signals in offline and online training. See Appendix B for data samples.

Table 1: Generalist and Expert Rubrics datasets statistics.

|  | **Train** | | **Eval.** | |
|---|---|---|---|---|
|  | # Sam. | # Rub. | # Sam. | # Rub. |
| Generalist | 1,500 | 15,528 | 487 | 5,003 |
| Expert | 1,864 | 33,554 | 332 | 5,938 |
| Math | 584 | 9,512 | 104 | 1,688 |
| Biology | 506 | 9,863 | 90 | 1,750 |
| Physics | 314 | 5,631 | 56 | 1,001 |
| Chemistry | 460 | 8,548 | 82 | 1,499 |

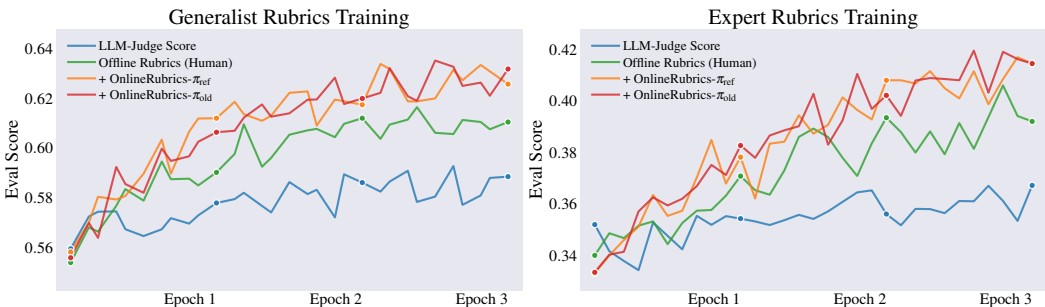

Figure 4: Results on the evaluation set of the Generalist and Expert datasets during training (higher is better). The evaluation set is fixed and does not contain any elicited rubrics. Both OnlineRubrics methods outperform using Offline Rubrics (Human) or LLM-judge Score (a Likert scale).

We evaluate OnlineRubrics on (1) evaluation sets of both datasets by calculating rubrics score and win rate using Gemini 2.5 Pro (Comanici et al., 2025)as an LLM-Judge, and (2) on the following public benchmarks: GPQA-Diamond (Rein et al., 2024), GSM8K (Cobbe et al., 2021), AlpacaEval (Li et al., 2023; Dubois et al., 2024)), and Arena-Hard (Li et al., 2024a;b).

## 6 EXPERIMENTS AND RESULTS

We begin by identifying the most effective LLM-based grader for rubric grading in Section 6.1. Next, we introduce our baselines in Section 6.2 and report the main results with OnlineRubrics in Section 6.3. Finally, we perform a qualitative analysis of the elicited rubrics in Section 6.4.

We train Qwen-2.5-7B-Instruct (Qwen et al., 2025) with GRPO as the training algorithm on the training data from both Generalist and Expert Rubrics datasets for 3 epochs and evaluate on the eval set of the respective datasets 10 times during each epoch. We use *o3-mini* as the LLM$_{extractor}$ and set the number of pairwise comparisons to 8. Appendix C provides the detailed experimental settings.

### 6.1 VERIFIER SELECTION

Rubrics training requires an LLM grader to evaluate whether an output $o_j$ meets the criteria specified in the rubrics $\mathcal{C}_i$. The input to the grader is a (prompt $x_i$, output $o_j$, rubrics $\mathcal{C}_i$) triplet, and the output is a sequence of binary scores indicating whether each criterion $c_k \in \mathcal{C}_i$ is satisfied by the output. Although grading is assumed to be easier than generation (Stechly et al., 2024), it is still a challenging task for LLMs and remains under-explored in previous work on rubrics due to the lack of human-annotated data with fine-grained rubric-level scores. However, different LLM graders have different evaluation capabilities, which can significantly affect the training of rubrics-based models. To address this, we have collected human evaluations of the original human-written rubrics for 2-6 sampled responses per prompt for 500 prompts for each of Expert and Generalist sets.

Using this dataset, we evaluate the performance of several LLM graders and present the results in Figure 3. Given that during the rubrics-based training, we need to evaluate multiple rollouts for each prompt, it is important to choose a grader with a low inference cost per sample. We calculate the inference cost per sample by dividing the total inference cost by the total number of samples.

Perhaps unsurprisingly, we find that all verifiers perform better on the Generalist dataset than the Expert dataset (average AUC score of 0.811 vs 0.768). Interestingly, the Pareto frontier for the Generalist dataset is the same as the Pareto frontier for the Expert dataset. This suggests that the relative performance of the verifiers is not affected by the domain.

### 6.2 BASELINES

We compare our methods with the following baselines:

**LLM-Judge Score** We train the model by only using an LLM judge to grade the responses on a Likert scale without any rubrics. The input to the LLM judge is a prompt-response pair $(x_i, o_j)$, and

| Model | Generalist Rub. | | Alpaca-Eval | | Arena-Hard |
|---|---|---|---|---|---|
| | Score | WR | WR | LC-WR | WR |
| *Baselines* | | | | | |
| Qwen-2.5-7B-Instruct | 55.4 | 39.0 | 30.0 | 28.2 | 50.0 |
| + LLM-Judge Score | 58.8 | 51.3 | 42.2 | 26.9 | 51.0 |
| + Offline Rubrics (Synthetic) | 58.8 | 52.8 | 39.5 | 28.2 | 51.5 |
| + Offline Rubrics (Human-written) | 61.0 | 62.2 | 46.4 | 28.0 | 52.4 |
| + Universal Requirements | 59.4 | 59.1 | 44.4 | 30.3 | 53.8 |
| + Pointwise Extraction | 62.9 | 64.9 | 48.1 | 29.4 | 51.1 |
| *Our Methods* | | | | | |
| + OnlineRubrics-$\pi_{\text{ref}}$ | 62.7 | 67.6 | 54.0 | **31.5** | 55.7 |
| + OnlineRubrics-$\pi_{\text{old}}$ | **63.2** | **68.2** | **55.0** | 30.4 | **56.5** |

Table 2: Results on the instruction-following benchmarks. WR stands for Win Rate and LC-WR is Length-Controlled Win Rate. We highlight the best performing model in each column in bold and underscore the second best performing approach. Both OnlineRubrics methods (OnlineRubrics-$\pi_{\text{ref}}$ and OnlineRubrics-$\pi_{\text{old}}$) are consistently better than the baselines except for one case.

| Model | Expert Rub. | | GPQA-D | GSM8K |
|---|---|---|---|---|
| | Score | WR | Acc. | Acc. |
| *Baselines* | | | | |
| Qwen-2.5-7B-Instruct | 33.6 | 31.9 | 34.7 | 79.2 |
| + LLM-Judge Score | 36.7 | 44.0 | 34.5 | 79.1 |
| + Offline Rubrics (Synthetic) | 37.1 | 46.4 | 36.6 | 79.2 |
| + Offline Rubrics (Human-written) | 39.2 | 51.8 | 36.2 | 79.9 |
| + Universal Requirements | 39.7 | 53.3 | 36.6 | 80.1 |
| + Pointwise Extraction | 40.9 | 57.1 | 33.6 | 78.3 |
| *Our Methods* | | | | |
| + OnlineRubrics-$\pi_{\text{ref}}$ | 41.4 | **61.0** | 37.6 | 80.0 |
| + OnlineRubrics-$\pi_{\text{old}}$ | **41.5** | 56.5 | **38.1** | **80.5** |

Table 3: Results on the professional-domain benchmarks. WR stands for win rate and Acc. stands for accuracy. We highlight the best performing model in each column in bold and underscore the second best performing approach. Both OnlineRubrics methods outperform the baselines.

the output is a Likert score that is converted to a reward $R_{i,j}$ using a linear mapping. We experiment with GPT-4.1-mini and o3-mini as the LLM judges. The prompt is given in Appendix D.

**Offline Rubrics (Synthetic)** We use the same prompts available in the Generalist and Expert Rubrics datasets. However, instead of using human-written rubrics, we synthetically create rubrics using o3-mini. See the prompt in Appendix D.

**Offline Rubrics (Human)** We train the model with human-written rubrics from the Generalist and Professional Rubrics datasets. As we shall see, this is better than using synthetic rubrics.

**Universal Requirements** As discussed in Section 2, previous work has argued that adding a fixed set of criteria to all samples helps the model to make training more stable and prevent reward hacking. We use the same universal requirements as in Viswanathan et al. (2025) and show OnlineRubrics, which elicits sample-grounded rubrics online, outperforms these universal requirements.

**Point-wise Elicitation** In order to show the effectiveness of pairwise comparison, we also extract rubrics point-wise using the same extractor model. The input to the extractor is prompt $x_i$, a response $o_j$ from the reference policy, and existing rubrics $\mathcal{C}_i$. The output is a set of criteria $C_i^e$ that we add to the human-written rubrics $\mathcal{C}_i$.

## 6.3 RESULTS AND DISCUSSION

Figure 4 shows the training curves for the Generalist and Expert datasets. Training with rubrics consistently scores higher and is more sample efficient than using LLM-Judge scores. More interestingly, adding the elicited rubrics during training (OnlineRubrics) improves the performance of the model on the evaluation sets of both datasets, which contain no elicited rubrics.

Table 2 and Table 3 present the results on a set of instruction-following and reasoning benchmarks, respectively. Training with Offline Rubrics (Human) is improving the performance of the model on all the respective datasets with the only exception being length controlled win rate on AlpacaEval (28.2% vs 26.9%). Importantly, training with Offline Rubrics (Human) is (a) always better than using LLM-Judge scores across all benchmarks, and (b) is better than using synthetic rubrics across 7 out of 9 evaluation metrics. More interestingly, adding the elicited rubrics to the offline rubrics (human-written) during training (OnlineRubrics) further boosts performance across both instruction-following and expert benchmarks. On AlpacaEval, for instance, OnlineRubrics-$\pi_{\text{ref}}$ increases the win rate from 46.4% to 55.0%, while also improving the length-controlled win rate (LC-WR) from 28.0% to 31.5% reflecting better quality responses in general.

When compared against other baselines, OnlineRubrics is consistently better than Universal Requirements across all benchmarks. This is interesting because it suggests that sample-grounded elicited rubrics are more effective than augmenting the rubrics with a set of fixed criteria that fail to capture the nuances of individual prompts and remain static as the policy evolves during training.

While adding pointwise extracted rubrics also often improves over offline rubrics, it is still surpassed by OnlineRubrics (48.1 vs. 54.0 and 55.0 on AlpacaEval, 51.1 vs. 55.7 and 56.5 on Arena-Hard). OnlineRubrics leverage pairwise differences to highlight discriminative properties that distinguish a better response from a worse one rather than relying on a single response.

## 6.4 QUALITATIVE ANALYSIS

We conducted a qualitative analysis of the elicited criteria and contrasted it with human-written rubrics. To summarize these changes, we applied an LLM-based comparison of rubric updates (between the initial rubrics and rubrics at the last epoch) followed by clustering to group recurring themes. We observe several consistent types of improvements have emerged. First, elicited criteria frequently introduced *evidence grounding (e.g., The response should include only categorically relevant, evidence-backed details.)*, *reproducibility (e.g., The response must avoid any process that can't be reproduced without modern technology.)*, and *holistic anti-gaming criteria (e.g, The response should avoid over-specification and over-enumeration.)*, broadening the evaluative focus beyond surface-level correctness. Second, many improvements emphasized *practicality and real-world feasibility*, adding criteria tied to implementation readiness and resource awareness. Third, we observed the addition of meta-criteria such as *structural organization*, *causal reasoning*, and *uncertainty handling*, which enhanced coverage of system-level and methodological dimensions.

Overall, the new criteria highlight that online elicitation tends to expand and strengthen rubrics over time. Instead of remaining fixed, criteria adapt dynamically as new errors or weaknesses are exposed, leading to more comprehensive and resilient evaluation standards. A complete list of clusters with proportions is reported in Appendix E.

## 7 CONCLUSION

We have described OnlineRubrics, a framework for dynamically eliciting new criteria from pairwise comparisons of responses during reinforcement learning. Unlike static rubrics which may be incomplete or become obsolete as training progresses, our approach aims to continuously surface overlooked errors or emerging desired properties. This yields robust gains across expert and generalist domains. Our results show improvements of up to 9 percentage points over training exclusively with human-written rubrics on AlpacaEval, GPQA and Arena-Hard. By moving rubric elicitation online, OnlineRubrics adapts as training evolves, capturing emergent behaviors and strengthening alignment beyond what fixed rubrics allow.

ETHICS STATEMENT

We have worked with human experts in creating our prompts and rubrics. All annotators are independent contractors and were compensated at rates consistent with fair labor practices and designed to align with applicable local laws. Participation to this study was entirely voluntary, with the option to decline tasks at any time. Importantly, we do not collect any personally identifiable information or sensitive data. Therefore, this work does not raise any ethical considerations.

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

## A  Proof for Proposition 1

*Proof.*

$$g_U - g_{R_t} = \mathbb{E}_{(x,o)}\Big[\nabla_\theta \log \pi_\theta(o|x)\big(U - R_t\big)\Big]$$

$$= \mathbb{E}_{(x,o)}\Big[\nabla_\theta \log \pi_\theta(o|x)\big(Y - \mathbb{E}_{(x,o)}\big[Y\big]\big)\Big] \qquad \text{where } Y = U - R_t$$

because $\mathbb{E}_{(x,o)}\big[\nabla_\theta \log \pi_\theta(o|x)\big] = 0$ we can center $Y$ without changing the expectation. Then

$$\left\| g_U - g_{R_t} \right\|_2 = \left\| \mathbb{E}_{(x,o)}\Big[\nabla_\theta \log \pi_\theta(o|x)\big(Y - \mathbb{E}_{(x,o)}\big[Y\big]\big)\Big] \right\|_2$$

$$\leq \sqrt{\mathbb{E}\Big[\big\|\nabla_\theta \log_{\pi_\theta}\big\|^2\Big]}\sqrt{Var(Y)} \qquad \text{by Cauchy-Schwarz}$$

$$= \sqrt{\mathbb{E}\Big[\big\|\nabla_\theta \log_{\pi_\theta}\big\|^2\Big]}\sqrt{Var(U - R_t)}$$

$$= \sqrt{\mathbb{E}\Big[\big\|\nabla_\theta \log_{\pi_\theta}\big\|^2\Big]}\sqrt{\mathbb{E}\big[(U - R_t)^2\big]}$$

$$= \sqrt{\mathbb{E}\Big[\big\|\nabla_\theta \log_{\pi_\theta}\big\|^2\Big]}\big\|w_I\big\|_1$$

$\square$

## B  DATA SAMPLES

We provide two samples showing sampled rollouts from current and reference policies, along with human and elicited rubrics in Figs. 5 and 6. Each criteria are preceded with its importance weight which range between 1-5 for Generalist and -10 and 10 for Expert sets.

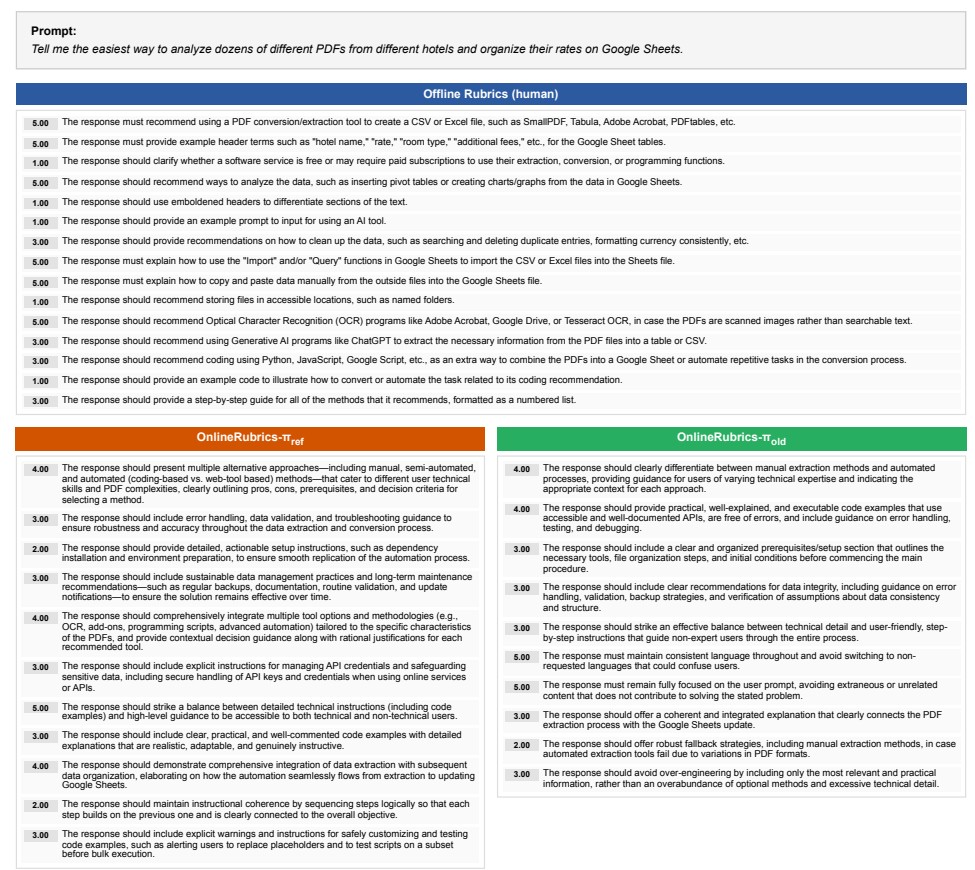

Figure 5: Data sample from the Generalist Rubrics dataset.

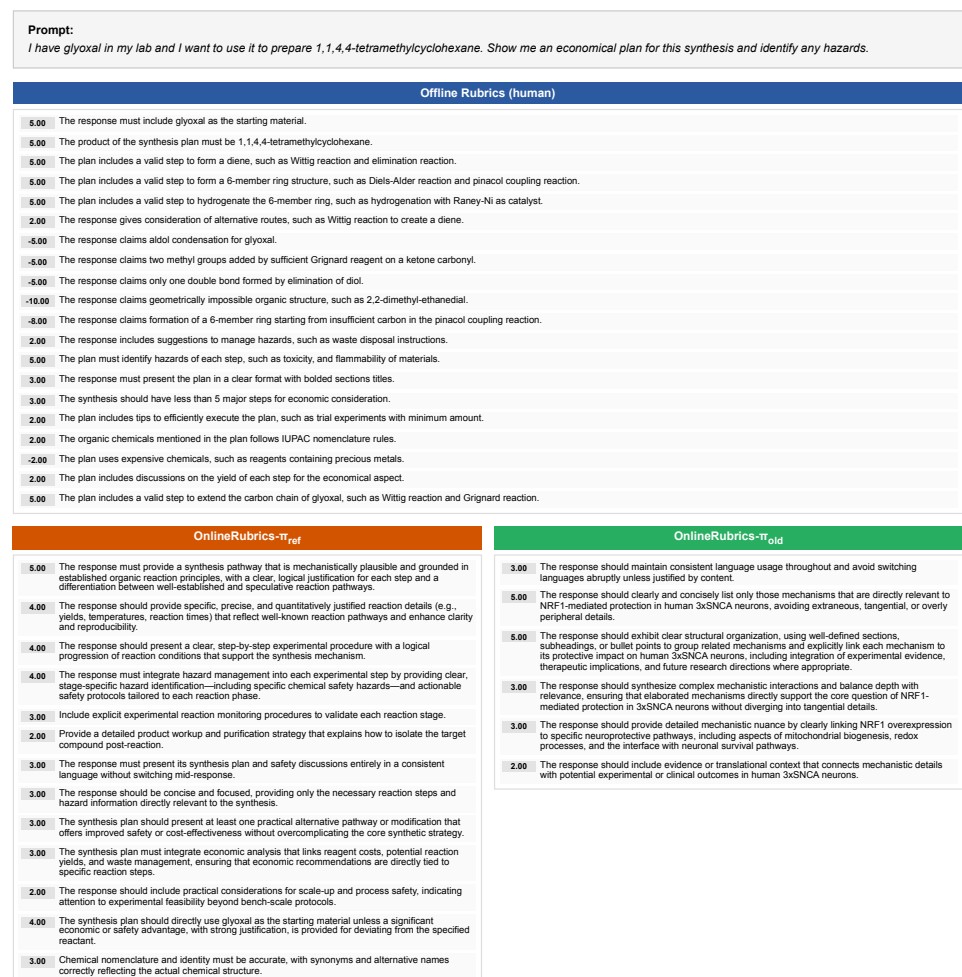

Figure 6: Data sample from the Expert Rubrics dataset.

## C    EXPERIMENTAL SETTINGS

### C.1    TRAINING SETTINGS.

We train Qwen-2.5-7B-Instruct (Qwen et al., 2025) on the training set of the Generalist and Expert Rubrics datasets for three epochs. Training follows the GRPO procedure described in Section 3, with 16 rollouts generated per sample. We use GPT-4.1-mini as the $LLM_{grader}$ and o3-mini as the $LLM_{extractor}$, performing eight pairwise comparisons per instance. Optimization uses a learning rate of $5e - 6$ with a warmup ratio of 0.1. KL-divergence regularization is applied with a coefficient of 0.01. All experiments are conducted on 8 NVIDIA H100 GPUs with per-device batch size of 6 and gradient accumulation of 2 steps (effective batch size is 96).

### C.2    EVALUATION SETTINGS.

**Generalist and Expert Rubrics Datasets.**    We calculate the score and win rate (vs. the reference policy) on the evaluation set of the Generalist and Expert Rubrics datasets. Score is calculated as explained in Eq. (4). We use GPT-4.1-mini as the $LLM_{grader}$. We use Gemini-2.5-Pro as the LLM-Judge that picks the winner between the two responses. For each sample, we run the judge twice by flipping the order of the two responses. If the judge picks the same response twice, we consider it as a win. The prompt for the judge is given in Appendix D.

| | |
|---|---|
| **Reproducibility & Transparency**
Transparent, stepwise reasoning with artifacts enabling independent reproducibility. | **8.96%** |
| **Practicality & Real-World Feasibility**
Criteria stressing implementation readiness, scalability, and real-world applicability. | **8.33%** |
| **Holistic Evaluation & Anti-Gaming**
Moving from checklists to holistic, anti-gaming principles emphasizing substance. | **7.69%** |
| **Lifecycle Management & Adaptivity**
Criteria supporting iterative feedback, adaptive management, and phase-based planning. | **7.42%** |
| **Structural Integrity & Organization**
Clear organization, modularity, and explicit information architecture in responses. | **6.58%** |
| **Mechanistic & Causal Reasoning**
Criteria requiring causal interpretability and validated mechanistic reasoning. | **6.23%** |
| **Method Selection & Justification**
Evidence-based justification and trade-off analysis of chosen methods. | **5.67%** |
| **Evidence-Based Reasoning & Provenance**
All claims grounded in verifiable evidence and explicit provenance, rejecting unsupported assertions. | **5.46%** |
| **Uncertainty, Robustness & Error Handling**
Explicit handling of uncertainty, edge cases, and error taxonomies. | **5.04%** |
| **Evidence Synthesis & Triangulation**
Integrating evidence across multiple methods and modalities for consistency. | **4.90%** |

Figure 7: Top-10 most frequent clusters of rubric criteria elicited via OnlineRubrics. Each cluster is shown with a short description and its share of samples, sorted by proportion.

**AlpacaEval.** We use the evaluation script[1] from (Li et al., 2023) to calculate the win rate and length controlled win rate on the evaluation set of AlpacaEval using the default settings.

**Arena-Hard.** We use the evaluation script[2] from (Li et al., 2024a;b) to calculate the win rate (vs. the reference policy) on the evaluation set of Arena-Hard. We use GPT-4.1 as the LLM-Judge.

**GPQA-Diamond.** We use simple-evals[3] for evaluation on GPQA-Diamond (Rein et al., 2024). We report the average accuracy across 4 runs.

**GSM8K.** We use lm-evaluation harness Gao et al. (2024) to calculate the strict match accuracy on the evaluation set of GSM8K (Cobbe et al., 2021).

## D  SYSTEM PROMPT TEMPLATES

Figures 8 and 9 show the system prompt templates used for $LLM_{extractor}$ and de-duplicating extracted criteria, respectively. We use the system prompt provided in Fig. 10 for $LLM_{grader}$.

Figures 11 and 12 show the system prompt templates used for LLM-Judge Score and LLM Judge for win rates, respectively. We use the system prompt provided in Fig. 13 to generate synthetic offline rubrics.

## E  QUALITATIVE RUBRIC CLUSTERS

We report the clusters of rubric refinements observed during online elicitation. Figure 7 lists each cluster with its name, a concise description, and its share of samples, sorted by proportion.

---

[1]https://github.com/tatsu-lab/alpaca_eval
[2]https://github.com/lmarena/arena-hard-auto
[3]https://github.com/openai/simple-evals

# F   USAGE OF LARGE LANGUAGE MODELS

We only use LLMs to aid or polish writing (e.g., grammar correction, style improvement, etc.) and carefully monitor the content to ensure it is accurate, not plagiarized, and not fabricated.

You are given a prompt and pair of responses to the same prompt. One of the responses is from a trained model and the other is from a baseline model.
Both responses are evaluated using an existing rubric. Your task is to identify their differences not already covered by the existing rubrics.
You should find the properties of one response that are better than the other.
Also, try to identify reward hacking patterns in the responses.
Reward hacking is a pattern where the response achieves a high score on rubrics by exploiting a loophole in the rubrics.
Think of reward hacking as a way to game the rubrics to get a high score. Reward hacking is like following the letter of the law but not the spirit of the law.

First, analyze both responses to identify the differences. Then, transform these observations into new evaluation criteria if they're not already covered by existing rubrics.
This is very important, any rubric that you introduce should be based on one of the responses.
Do not use your own knowledge to introduce new criteria that are not based on one of the responses.
Focus on criteria that distinguish genuinely helpful responses from those gaming the system. Also, keep an eye out for language switching patterns that might confuse the verifier.
Make sure the new criteria follow the same style as the existing criteria.
Assign a positive weight (integer) to each of the new criteria based on the relative importance of the criterion to the existing criteria.

Output format:
```json
{
  "analysis": "Your analysis of reward hacking patterns in the responses and good/bad behaviors that should be encouraged/discouraged. It's okay for the analysis to be long.",
  "new_criteria": [
    {
      "quote": "quote from the response following/violating the criterion",
      "criterion": "criterion_text",
      "weight": criterion_weight
    }
  ]
}
```

If no meaningful new criteria are needed, output:
```json
{
  "analysis": "Your analysis...",
  "new_criteria": []
}
```

Figure 8: Full system prompt template used for LLM$_{\text{extractor}}$.

You will review a collection of candidate evaluation criteria from multiple response comparisons and remove redundancy while preserving the best unique criteria. Your goal is ONLY to deduplicate and aggregate, NOT to introduce new criteria or remove criteria entirely.

## Your Task: Deduplication and Aggregation ONLY

You should:
- **Remove redundant/overlapping criteria** that say essentially the same thing
- **Merge similar criteria** by combining them into a single, clearer criterion
- **Aggregate weights** for merged criteria (e.g., if two similar criteria have weights 3.0 and 4.0, the merged criterion might get weight 3 or 4).
- **Preserve all unique criteria** that address different quality aspects
- **Keep the original wording** when possible, only clarifying when necessary

You should NOT:
- **Add completely new criteria** not present in the candidate list
- **Remove criteria entirely** unless they are truly redundant
- **Change the intent** of existing criteria
- **Introduce your own knowledge** beyond what's in the candidates

## Deduplication Process

1. **Group similar criteria** - Identify candidates that address the same quality aspect
2. **Select best wording** - Choose the clearest, most specific wording from each group
3. **Aggregate weights** - Combine weights from merged criteria appropriately. Only use positive integers.
4. **Preserve unique criteria** - Keep all criteria that address different aspects
5. **Maintain quality focus** - Ensure the final set covers all important quality dimensions from candidates

## CRITICAL: You MUST end your response with JSON

```json
{
  "analysis": "Your analysis of redundancy patterns and merging decisions...",
  "final_criteria": [
    {
      "criterion": "Deduplicated criterion text (merged from similar candidates)",
      "weight": criterion_weight
    }
  ]
}
```

If all criteria are unique (no deduplication needed), return all candidates:

```json
{
  "analysis": "No redundancy found, all criteria are unique...",
  "final_criteria": [
    {
      "criterion": "Original candidate criterion 1",
      "weight": criterion_weight
    },
    {
      "criterion": "Original candidate criterion 2",
      "weight": criterion_weight
    }
  ]
}
```

Figure 9: Full system prompt template used for de-duplicating extracted criteria.

You are an expert evaluator. Given a user prompt, a generated response, and a list of quality rubrics, please evaluate the response against EACH rubric.

For each rubric,
- Mark "PRESENT" if the criterion is satisfied, or "NOT_PRESENT" if it is not. For example, given the response "Apples are red", the rubric "Mentions apples" is PRESENT, "Does not mention strawberries" is also PRESENT since the response doesn't mention strawberries and "Mentions oranges" is NOT_PRESENT. Also, "Avoids mentioning strawberries" is PRESENT because the response doesn't mention strawberries. However, "Avoids mentioning apples" is NOT_PRESENT because the response mentions apples.
- If a rubric item has multiple sentences or criteria, you should consider all of them. If any of the criteria is not met, the answer should be NOT_PRESENT. Only return PRESENT if all of the criteria are met.
- One important exception to the above bullet point is that if a rubric says "such as", "for example", or "including", the response does not have to include all of the examples listed to meet the criteria. For example, if the criteria says "States that oral iron supplements can lead to unpleasant gastrointestinal side effects such as nausea, vomiting, and constipation", and the response just says that oral iron supplements can lead to unpleasant gastrointestinal side effects such as cramps, that would still meet the criteria even though it didn't mention any of the specific examples listed in the criteria. That is, there are no partial credit for any of the criteria.

Start your response with a valid JSON object that starts with "```json" and ends with "```".

The keys must be the numbers of the rubrics provided and the values must be either "PRESENT" or "NOT_PRESENT" based on your evaluation. Ensure the JSON is valid and contains no extra text or explanations.

Example response:
```json
{
 "1": "PRESENT",
 "2": "NOT PRESENT",
 "3": "PRESENT"
}
```

Figure 10: Full system prompt template used for LLM$_{\text{grader}}$.

You are an expert evaluator. Given a user prompt and a generated response, please rate the overall quality of the response on a scale of 1 to 10, where 1 is very poor and 10 is excellent.

Start your response with a valid JSON object that starts with "```json" and ends with "```".
The JSON object should contain a single key "rating" and the value should be an integer between 1 and 10.

Example response:
```json
{
 "rating": 8
}
```

Figure 11: Full system prompt template used for LLM-Judge Score.

Please act as an impartial judge and evaluate the quality of the responses provided by two AI assistants to the user question displayed below.
You should choose the assistant that follows the user's instructions and answers the user's question better.
Your evaluation should consider factors such as the helpfulness, relevance, accuracy, depth, creativity, and level of detail of their responses.

Begin your evaluation by comparing the two responses and provide a short explanation.
Avoid any position biases … Do not allow the length of the responses to influence your decision.
After providing your explanation, output your final verdict by strictly following this format:

"[[A]]" if assistant A is better, "[[B]]" if assistant B is better, and "[[C]]" for a tie.

Figure 12: Full system prompt template used for LLM Judge for win rates.

Your job is to generate a self-contained set of evaluation criteria ("rubrics") for judging how good a response is to a given question.

Terminology:
- A prompt is a task description (question) that a user gives to a model.
- A response is a model's output when given the prompt.
- A rubric is a set of criteria that capture the elements of an ideal response given a prompt. The rubric will be used to evaluate the quality of a response to the prompt. Rubrics can cover aspects of a response such as, but not limited to, factual correctness, ideal-response characteristics, style, completeness, helpfulness, harmlessness, patient-centeredness, depth of reasoning, contextual relevance, and empathy.
- A criterion is a single item in a rubric.

A good rubric follows these principles:
- As a whole, the rubric should be Mutually Exclusive (avoid overlapping criteria) and Collectively Exhaustive (all requests of the prompt should be covered).
- Each item should test one idea. If an item tests for the presence of X and Y, it should be split into two items (unless no reasonable prompt would contain one without the other).
- Each item should be binary (have yes/no answers) and as objective as possible. :x: "Response is too verbose" → :white_check_mark: "Response is less than 500 words long"
- Each item should be self-contained and include sufficient detail so that an uninformed grader can verify it without external knowledge. E.g. :x: "Names a 2010 Nobel Prize winner" → :white_check_mark: "Identifies one of the following 2010 Nobel Prize winners: A, B, or C".
- Avoid criteria that doesn't allow for partial credit. E.g. :x: "Mentions 3 Nobel Prize winners A, B, and C" → Split into "Mentions Nobel Prize winner A", "Mentions Nobel Prize winner B", "Mentions Nobel Prize winner C". All these should be detailed enough so that an uninformed grader can verify them without external knowledge.

Also consider the following axes when helping the user to improve the rubric:
- Communication Quality: Response length, clarity, level of detail, vocabulary, and structure are well-matched to the user and situation.
- Instruction Following: Adheres to the user's directions for how to complete the task or how to format a response. Satisfies all user constraints and answers all questions.
- Accuracy: Includes only factually correct information. Information is supported by evidence or consensus and uncertainty is expressed when evidence is limited.
- Context Awareness: Responds appropriately given the user's context (e.g., user role, setting, resources) and seeks clarification when needed.
- Completeness: Addresses all parts of the query needed for a safe and helpful response. Even if accurate, a response that omits key steps or considerations can still result in low-quality advice or harm.

Your task it to generate criteria for a given prompt. Also, you should assign a weight to each criterion. Weights should be an integer between 1 and 10.
Your response should be a json object with the following format. Use the reasoning fields to think about the criteria and reason about it.
```
{
    "initial_reasoning": INITIAL_REASONING,
    "rubrics": [
        {
            "reasoning": REASONING,
            "criterion": CRITERION,
            "weight": WEIGHT,
        },
    ]
}
```

Figure 13: Full system prompt template used to generate synthetic rubrics.

