# OpenReview forum: "Online Rubrics Elicitation from Pairwise Comparisons"
_ICLR.cc/2026/Conference — Submitted to ICLR 2026_

### Official Review · Reviewer_wkK2 · 2025-10-26

**Soundness:** 3
**Presentation:** 4
**Contribution:** 4
**Rating:** 8
**Confidence:** 4

**Summary:**

In this paper, the authors argue that offline rubrics (static, pre-defined rubrics) are often incomplete for RL in training LLMs. For example, they cannot adapt to emergent model behaviors like self-praising, and they are vulnerable to reward hacking. To overcome this, they introduce OnlineRubrics, a framework that dynamically elicits new evaluation criteria during training. The core mechanism involves:

1. At each training step, generating a pair of responses to a prompt: one from the current policy and one from a control policy (a reference model or the previous iteration of the policy).
2. Using an LLM extractor to perform a pairwise comparison of these responses, identify meaningful differences not covered by the existing rubric, and formulate these differences as new, weighted criteria.
3. Augmenting the original rubric with these new criteria to compute a more comprehensive reward signal for the policy update.

The authors empirically demonstrate that OnlineRubrics improves performance by up to 8% relative to baseline using offline rubrics across both generalist (AlpacaEval, Arena-Hard) and expert-domain (GPQA, GSM8K) benchmarks. A qualitative analysis reveals that the elicited rubrics tend to focus on important themes such as evidence grounding, reproducibility, and anti-gaming.

**Strengths:**

* **Originality**: The paper's originality is outstanding and represents its primary strength. It improves LLM alignment by moving reward generation from a static, offline process to a dynamic, online one. The idea of using pairwise comparisons between the current and a control policy during training to elicit new, emergent evaluation criteria is new to me. This creates a self-correcting loop in which the reward model evolves alongside the policy, a departure from existing RLHF/RLAIF frameworks.
* **Quality**: The research demonstrates empirical results across diverse, challenging benchmarks, including AlpacaEval, Arena-Hard, GPQA, and GSM8K. The fact that the method yields gains in both generalist and expert domains speaks to its robustness. The inclusion of multiple strong baselines (human-written rubrics, pointwise extraction) and insightful ablations provides a rigorous validation of the framework's effectiveness.
* **Clarity**: The core mechanism illustrated in Figure 1 provides a simple, intuitive walkthrough of the iterative process. The motivation is well established, and the results are presented in an easy-to-interpret way.
* **Significance**: This work is significant as it offers a promising solution to the critical problem of reward hacking and brittle reward models in LLM alignment. Static rewards are a known limitation of current RL methods. By proposing a way to make the reward function adaptive, this paper provides a practical technique that could lead to more robust, capable, and well-aligned models. The qualitative analysis of the elicited rubrics also provides significant insight into LLM failure modes and how to address them.

**Weaknesses:**

* **Dependence on the Extractor LLM**: The framework's performance is bottlenecked by the quality of the extractor LLM. It could degrade or misguide the training if the extractor fails to identify key differences, generates noisy rubrics, or is itself biased. A more thorough discussion of the component's potential failure modes and robustness would strengthen the paper.
* **Computational Complexity**: The proposed method introduces an additional & heavyweight LLM extractor call inside the training loop for every batch of data. This could be computationally more expensive than training with an offline rubric. An analysis of the computational overhead would be a welcome addition.
* **Scalability of Rubrics**: It looks like the rubric for each prompt tends to grow over time. It's unclear how the system handles an ever-expanding list of criteria. This may slow down overall training to an unbearable pace, or the rubrics may become too complex for another LLM to follow.

**Questions:**

1. How robust is the rubric elicitation process to the choice of the extractor model? Do you experiment with weaker models than o3-mini? Is there a performance threshold below which the extractor starts to introduce more noise than signal?
2. Could you provide an analysis of the computational overhead introduced by the online elicitation step compared to the baseline of training with offline rubrics? What is the increase in training time in practice?
3. A key justification for your design is that pointwise elicitation is inferior to the pairwise approach. Could you elaborate on why you believe this is the case? Is it because the pairwise comparison naturally focuses the extractor model on the most salient differences, which are more likely to be useful as discriminative criteria? Or are there other reasons?

---

> ### Author Response · Authors · 2025-11-24
> **Official Comment by Authors**
>
> We thank you for your review and constructive feedback. It is encouraging to hear that you found our work significant and outstanding in its originality and empirical results. Please find our responses to your concerns below.

---

> > ### Author Response · Authors · 2025-11-24
> > **Computational Complexity Overhead (W2)**
> >
> > Thank you for raising this important point, we believe this is a valuable addition to our discussion.
> >
> > We would like to clarify that the computational cost overhead of OnlineRubrics primarily arises from LLM calls for rubric extraction and pairwise comparisons, not generation of the responses themselves. $\pi_\text{old}$ uses the rollouts generated by the policy for GRPO, and $\pi_\text{ref}$ uses rollouts from step 0 of GRPO, hence there is no additional cost in generating the responses.
> >
> > We acknowledge that there's a computational cost associated with: (1) rubric elicication and (2) grading with a longer rubric. Yet, this cost is within the same order of magnitude as verifying responses using LLM-based verifiers, which is a common practice in rubrics training [1, 2]. At each GRPO iteration, scoring responses against rubrics requires an LLM-judge to grade each response once. Let's denote the number of responses as N (we use N=16 rollouts in our experiments). Therefore, the cost of scoring responses is O(N).
> >
> > In OnlineRubrics, we use a pair of responses for each comparison, and we perform K comparisons per GRPO iteration (K=8 in our default setting). Thus, the cost of comparisons is O(2K). Also, note that by design $K \leq N$. Therefore, although there are 8+1 (8 comparisons + 1 final processing for deduplication) LLM calls per GRPO iteration in OnlineRubrics, the overall computational cost remains $O(N)$, which is within the same order of magnitude as traditional LLM-based verification methods.
> >
> > Note that the costs arising from (2) is negligible as the LLM grader evaluates all criteria in a single pass. The number of new extracted criteria are usually less than the existing criteria and as a whole, they are much shorter than the response itself, which is the majority of the computation time.
> >
> > In practice, a single baseline training costs $\approx$ 100 USD for rubric-based verification and $\approx$ 150 USD for OnlineRubrics with K=8 comparisons, indicating a 1.5x overhead. We believe this is a reasonable trade-off given the significant performance improvements demonstrated by OnlineRubrics.
> >
> >
> >
> > [1] Gunjal, A., Wang, A., Lau, E., Nath, V., He, Y., Liu, B., & Hendryx, S. (2025). Rubrics as Rewards: Reinforcement Learning Beyond Verifiable Domains. arXiv:2507.17746.
> >
> > [2] Viswanathan, V., Sun, Y., Ma, S., Kong, X., Cao, M., Neubig, G., & Wu, T. (2025). Checklists Are Better Than Reward Models for Aligning Language Models. arXiv:2507.18624.

---

> > ### Author Response · Authors · 2025-11-24
> > **Scalability of Rubrics (W3)**
> >
> > We would like to clarify that the rubrics do not grow over time. At each training step, OnlineRubrics elicits a fresh set of on-policy rubrics based on the current policy's behavior. Thus, there is no risk of unbounded growth in the number of criteria or accumulation of noise from previous epochs.

---

> > ### Author Response · Authors · 2025-11-24
> > **Pointwise vs Pairwise Elicitation (Q3)**
> >
> > As discussed in Section 4, we believe that pairwise elicitation helps the extractor find potential errors and identify useful features in either of the two responses. Additionally, pairwise comparison mitigates the risk of the extractor generating generic criteria or introducing its own biases, as it must focus on the specific differences between the two responses provided. This is evident in the results shown in Tables 2 and 3, where OnlineRubrics consistently outperforms the pointwise baseline.

---

> ### Author Response · Authors · 2025-11-24
> **Dependence on the Extractor LLM (W1 + Q1)**
>
> We have experimented with various extractor models, including stronger models like `gpt-4.1` and `gpt-4.1-mini`, as well as an open-source model `gpt-oss-120B`. The new results table below summarize the performance of OnlineRubrics with these different extractors. The results show that while stronger extractors (o3-mini, GPT-4.1) tend to yield better performance; smaller or open-source models still provide consistent improvements over optimizing against and LLM-Judge and `gpt-oss-120B` as an extractor results in competitive performance in Alpaca eval.
>
> | Model | Extractor | Alpaca-Eval WR | Alpaca-Eval LC-WR | Generalist Score |
> | -------|-----------|-----------------|--------------------|------------------|
> | Qwen-2.5-7B-Instruct | - | 30.0 | 28.2 | 55.4 |
> | &nbsp;&nbsp;+ LLM-Judge Score | - | 42.2 | 26.9 | 58.5 |
> | &nbsp;&nbsp;+ Offline Rubrics (Human) | - | 46.4 | 28.0 | 61.0 |
> | &nbsp;&nbsp;&nbsp;&nbsp;+ OnlineRubrics-$\pi_\text{ref}$ | o3-mini | 54.0 | 31.5 | 62.7 |
> |                                                          | gpt-4.1 | 49.9 | 25.4 | 61.9 |
> |                                                          | gpt-4.1-mini | 48.9 | 28.0 | 61.8 |
> |                                                          | gpt-4.1-nano | 50.5 | 27.7 | 60.4 |
> |                                                          | gpt-oss-120B | 43.1 | 30.2 | 60.4 |
>
> Through manual inspection, we have identified several potential failure modes of OnlineRubrics that could be addressed in future work. In some cases, the extracted criteria were challenging to grade. For example, a criterion like *"The response explains why the distinction is necessary"* can be subjective and hard to evaluate if the grader lacks sufficient context or expertise. This could lead to inconsistent scoring and potentially misguide the policy updates.
>
> Another failure mode we observed is that some of the extracted criteria were not atomic enough, leading to ambiguity in grading. For example, a criterion such as *"The response should be concice and provide clear explanations without unnecessary jargon."* combines multiple aspects (conciseness and clarity) into a single criterion, which can make it difficult to assess whether a response meets both requirements adequately.
>
> We believe that addressing these failure modes could further enhance the effectiveness of OnlineRubrics. Future work could explore methods to refine the rubric extraction process, including post-training the extractor, to ensure that criteria meet standards we had for human-written rubrics.

---

### Official Review · Reviewer_yBS8 · 2025-10-30

**Soundness:** 3
**Presentation:** 4
**Contribution:** 3
**Rating:** 6
**Confidence:** 3

**Summary:**

This work proposes a new approach to generating rubric-based reward signals in LLM post-training. Instead using pre-configured rubrics per prompt, the method uses "online" rubrics that are dynamically generated based on the current model capabilities. The rubrics are per prompt rather than for the entire dataset. They evaluate their method across a number of datasets, including AlpacaEval, GPQA and ArenaHard. The authors report impressive results indicating that their online rubrics based approach is indeed able to improve performance (in terms of win-rate) across this diverse set of benchmarks.

**Strengths:**

1. **Clearly written.** Overall the quality of the writing is high, although some sentences are a bit long.
2. **Strong motivation.** The authors outline clearly why online rubrics are preferred over fixed ("offline") rubrics.
3. **Strong experimental results.** On the diverse set of experiments tested by the authors, they are able to show impressive performance improvements.

**Weaknesses:**

1. **Only single-seed experiments reported.** Since all training methods have some non-deterministic part would be interesting to see mean/var reported over, e.g., 3 seeds per method. Such an extension would allow a better understanding of the overall method.


*Minor (no impact on score, no need to respond):*
1. L039: I found the example of a rubric from a finance context confusing, I think you could select a more quickly understandable one.
2. L054: Font size in Fig 1 is quite small, had to read. Maybe some text could be removed to make space
3. L210: I don't think this example is very good, seems like both responses are fine: one considers that caltrains don't run all night (thus returning by car may be favoured)? I think there should be a clearer example.
4. L251: missing "in" for "*in* Section 5".

**Questions:**

1. Have you considered validating the rubrics experimentally beyond de-duplication before training on them? Related work [1] on interpreting pairwise feedback datasets extracts rubrics (referred to as "principles") but also validates the rubrics ability to reconstruct feedback. A similar approach could be used to simply check if a generated rubric is actually able to distinguish responses, as this does not appear to be done at the moment (the only check is de-duplication)?
2. Are you planning to release the datasets of (1) human evaluations for human-written rubrics discussed in Section 6.1 and (2) the generalist and expert rubrics datasets collected for evaluation? I was unable to find a corresponding link.

[1] Findeis, Arduin, et al. "Inverse Constitutional AI: Compressing Preferences into Principles." _The Thirteenth International Conference on Learning Representations_ (2025).

---

> ### Author Response · Authors · 2025-11-24
> **Official Comment by Authors**
>
> We thank you for your thoughtful review. We are encouraged that you found the motivation strong, empirical results convincing and paper well-written. We will fix the minor issues and typos you pointed out. Please find our responses to your concerns below.

---

> > ### Author Response · Authors · 2025-11-24
> > **Single-Seed Experiments (W1)**
> >
> > We agree that multiple seeds would provide a better understanding of variance. However, due to the resource constraints and computational cost of our experiments, we decided to focus on a single seed for the main experiments to allow us to explore more configurations and baselines.
> >
> > It is worth pointing out that the training curves in Figure 4 demonstrate consistent improvements of OnlineRubrics over baselines throughout the training process, suggesting that the observed gains are robust rather than due to random chance.
> >
> > Following the suggestions of other reviewers, we're glad to report additional results with different base models in the new results tables above, which further supports the generality of our findings.
> >
> > | Model | Alpaca-Eval WR | Alpaca-Eval LC-WR | Generalist Rubrics |
> > |-------|-----------------|--------------------|------------------|
> > | Llama-3.2-3B-Instruct | 13.5 | 12.6 | 45.9 |
> > | &nbsp;&nbsp;+ LLM-Judge Score | 18.2 | 14.4 | 50.5 |
> > | &nbsp;&nbsp;+ Offline Rubrics (Human) | 19.4 | 14.4 | 51.6 |
> > | &nbsp;&nbsp;&nbsp;&nbsp;+ OnlineRubrics-$\pi_\text{old}$ | 20.6 | 16.2 | 52.0 |
> >
> > | Model | GPQA-D Acc. | GSM8K Acc. | Expert Rubrics |
> > |-------|-------------|------------|--------------|
> > | Llama-3.1-8B-Instruct | 27.2 | 56.7 | 25.1 |
> > | &nbsp;&nbsp;+ LLM-Judge Score | 31.7 | 62.4 | 34.3 |
> > | &nbsp;&nbsp;+ Offline Rubrics (Human) | 28.7 | 59.3 | 34.9 |
> > | &nbsp;&nbsp;&nbsp;&nbsp;+ OnlineRubrics-$\pi_\text{old}$ | 30.0 | 60.6 | 36.5 |
> >
> > We ran additional experiments using ``Llama-3.2-3B-Instruct`` with generalist and ``Llama-3.1-8B-Instruct`` with expert domains. The results, included in the new results tables above, show that OnlineRubrics continues to provide significant improvements over baselines across different model families and sizes. This suggests that our method is not model-specific and can generalize well to other LLMs.
> >
> > Please note that due to time and resource constraints, we have only tested OnlineRubrics-$\pi_\text{ref}$ on these additional models, which as discussed in the paper is a strong variant of our method.

---

> > ### Author Response · Authors · 2025-11-24
> > **Questions**
> >
> > ### De-duplication Validation (Q1)
> >
> > We appreciate this suggestion. While we have not conducted a formal validation of the extracted rubrics' ability to reconstruct feedback, we did perform manual inspections of a sample of extracted rubrics to ensure their relevance and quality. These inspections confirmed that the extracted rubrics were generally effective in distinguishing between different responses, aligning with the intended purpose of guiding policy improvement.
> >
> > Additionally, we would like to point out that by design, if a rubric fails to distinguish between the two responses in a comparison, it would not contribute to the reward signal during training when calculating the advantage. Thus, ineffective rubrics would naturally have a diminished impact on the policy updates. However, the results in Tables 2 and 3 demonstrate that OnlineRubrics consistently outperforms baselines, indicating that the extracted rubrics are indeed effective in guiding policy improvement.
> >
> > ### Dataset Release (Q2)
> >
> > Thank you for asking about the data. We are currently in the process of clearing legal reviews with the aim of releasing more data to the community soon. In the meantime, we have included samples in Appendix B.

---

### Official Review · Reviewer_nonc · 2025-10-31

**Soundness:** 3
**Presentation:** 3
**Contribution:** 3
**Rating:** 6
**Confidence:** 4

**Summary:**

Introduces dynamic rubric (as opposed to current static rubric setups) as a reward during reinforcement learning. This reduces the chance of reward hacking while training. OnlineRubrics is a framework that uses a response from a control model to update the policy while training. Final performance is improved over static rubrics (whether human or synthetic).

**Strengths:**

1. Nice simple proposal, easy to reason about and well explained.
2. Great that all system prompts/instruction prompts have been shared and various models have been compared in performance.
3. Reasonable improvement, and ablating various choices.

**Weaknesses:**

1. Not 100% clear whether the performance improvement is because of a better final rubric, or because it's dynamic during training. E.g. no ablation on whether the frequency of updating the rubric matters much, or if we use the final updated rubric and train on that from scratch. We know the human rubric outperforms the synthetic rubric by quite a bit, while it's still static, are we not creating a new rubric that is just even better (but could have been static)?
2. No real analysis on reward hacking after using the dynamic rubric framework. You do mention that it reduces the chance of reward hacking, and the assumption feels right, but there is no evidence (could even be through manual inspection) that this happens.
3. Hard to answer, but I do not see anything about what it still does wrong. I can imagine it might go into too many details, or move away from the original policy, but it would be great for future researchers to have better insight in where this framework is still failing (even though we see result improvements).
4. Would have been nice to see some smaller open-source models as judges (e.g. Qwen, or others) such that we don't rely on private APIs too much.

**Questions:**

1. What happens if the control model is a small/cheap model?
2. What if you update the policy using only a subset of the batch? E.g. only 2 items per batch? Does that hurt the performance much?
3. What if you use the final rubric that was dynamically updated in a new training? Are we mainly seeing improvements because we have a better rubric (similar to synthetic vs human), or because it’s actually dynamic during training?

---

> ### Author Response · Authors · 2025-11-24
> **Official Comment by Authors**
>
> We thank you for your positive review and constructive feedback.  We are glad that you found OnlineRubrics to be simple yet with reasonable improvements and well-explained. Please find our responses to your concerns below.

---

> > ### Author Response · Authors · 2025-11-24
> > **Impact of Dynamic Rubric Extraction (W1 + Q3)**
> >
> > Thank you for this insightful question. We would like to note that we have two baselines that add static rubrics to the human-written rubrics: (1) Univeral Requirements (a set of static criteria added to all prompts) and (2) Pointwise Extraction (extracting rubrics for each prompt once at the start of training based on the responses from the initial policy $\pi_\text{ref}$). OnlineRubrics outperforms both of these static baselines, indicating that the dynamic nature of rubric extraction during training is a key contributor to the performance gains.
> >
> > We believe this question relates to the broader distinction between on-policy and off-policy learning. Extracting all rubrics up front and then training only on that fixed set would effectively become off-policy learning and would inherit its well-known theoretical limitations.

---

> > ### Author Response · Authors · 2025-11-24
> > **Reward Hacking Analysis (W2)**
> >
> > Thank you for raising this important point. We manually inspected a sample of extracted criteria and identified several instances where the criteria directly address potential reward hacking behaviors.
> >
> > For example, in one case, an extracted criterion was *"The response should provide region-specific guidance by referencing official materials, correct regulatory authorities (e.g., DVTA for Northern Ireland), and offering accurate, consistent descriptions of the exam format, structure, and administrative requirements."*  where the prompt was *Give me advice on how to pass the theory exam for minibuses (D1) in Northern Ireland.*
> > This criterion was extracted when one of the responses was providing generic advice that did not specifically address the Northern Ireland context, which could be seen as a form of reward hacking by providing superficially correct but ultimately unhelpful information (*misgeneralization of context*).
> >
> > In another instance, the criterion *"The response should focus on the specified key topics (hazard identification, risk assessment, incident reporting, and worker participation) by avoiding extraneous details and ensuring each element is addressed with sufficient depth."* was extracted for a prompt about occupational health and safety management. This criterion helped mitigate reward hacking where an on-policy response included irrelevant information to appear comprehensive without actually addressing the key topics.
> >
> > There were also numerous new criteria that focused on "instruction following", "diversity", and "conciseness", which are common dimensions where reward hacking can occur. These examples illustrate how OnlineRubrics can dynamically identify and address potential reward hacking behaviors through its rubric extraction process.

---

> > ### Author Response · Authors · 2025-11-24
> > **Failure Modes and Future Work (W4)**
> >
> > Through manual inspection, we have identified several potential failure modes of OnlineRubrics that could be addressed in future work. In some cases, the extracted criteria were challenging to grade. For example, a criterion like *"The response explains why the distinction is necessary"* can be subjective and hard to evaluate if the grader lacks sufficient context or expertise. This could lead to inconsistent scoring and potentially misguide the policy updates.
> >
> > Another failure mode we observed is that some of the extracted criteria were not atomic enough, leading to ambiguity in grading. For example, a criterion such as *"The response should be concice and provide clear explanations without unnecessary jargon."* combines multiple aspects (conciseness and clarity) into a single criterion, which can make it difficult to assess whether a response meets both requirements adequately.
> >
> > We believe that addressing these failure modes could further enhance the effectiveness of OnlineRubrics. Future work could explore methods to refine the rubric extraction process, including post-training the extractor, to ensure that criteria meet standards we had for human-written rubrics.

---

> > ### Author Response · Authors · 2025-11-24
> > **Smaller Open-Source Judges (W5)**
> >
> > Please see Section 6.1 (Verifier Selection) where we experiment with multiple verifiers of varying sizes, including open-source models like `llama-4` and analyze their cost-performance trade-offs.
> > We intentionally used a relatively strong verifier (`GPT-4.1-mini`) that balances performance and cost to ensure reliable scoring to isolate the effects of verification from other components of our method (i.e., rubric extraction).

---

> > ### Author Response · Authors · 2025-11-24
> > **Smaller/Cheaper Extractors (Q1)**
> >
> > We have experimented with various extractor models, including stronger models like `gpt-4.1` and `gpt-4.1-mini`, as well as an open-source model `gpt-oss-120B`. The new results tables below summarize the performance of OnlineRubrics with these different extractors. The results show that while stronger extractors (o3-mini, GPT-4.1) tend to yield better performance; smaller or open-source models still provide consistent improvements over optimizing against and LLM-Judge and `gpt-oss-120B` as an extractor results in competitive performance in Alpaca eval.
> >
> > | Model | Extractor | Alpaca-Eval WR | Alpaca-Eval LC-WR | Generalist Score |
> > | -------|-----------|-----------------|--------------------|------------------|
> > | Qwen-2.5-7B-Instruct | - | 30.0 | 28.2 | 55.4 |
> > | &nbsp;&nbsp;+ LLM-Judge Score | - | 42.2 | 26.9 | 58.5 |
> > | &nbsp;&nbsp;+ Offline Rubrics (Human) | - | 46.4 | 28.0 | 61.0 |
> > | &nbsp;&nbsp;&nbsp;&nbsp;+ OnlineRubrics-$\pi_\text{ref}$ | o3-mini | 54.0 | 31.5 | 62.7 |
> > |                                                          | gpt-4.1 | 49.9 | 25.4 | 61.9 |
> > |                                                          | gpt-4.1-mini | 48.9 | 28.0 | 61.8 |
> > |                                                          | gpt-4.1-nano | 50.5 | 27.7 | 60.4 |
> > |                                                          | gpt-oss-120B | 43.1 | 30.2 | 60.4 |

---

> > ### Author Response · Authors · 2025-11-24
> > **Policy Update with Subset of Batch (Q2)**
> >
> > Thanks for this interesting question. We have conducted ablation studies varying the number of comparisons used in OnlineRubrics. The new results tables below include performance metrics for 1, 2, 4, and 8 comparisons. The results indicate that while performance generally improves with more comparisons, even a single comparison still yields notable improvements over the baselines. This demonstrates the robustness of OnlineRubrics to the number of comparisons used.
> >
> > | Model | \# Comparisons | Alpaca-Eval WR | Alpaca-Eval LC-WR | Generalist Score |
> > |-------|----------------|-----------------|--------------------|------------------|
> > Qwen-2.5-7B-Instruct | - | 30.0 | 28.2 | 55.4 |
> > | &nbsp;&nbsp;+ LLM-Judge Score | - | 42.2 | 26.9 | 58.5 |
> > | &nbsp;&nbsp;+ Offline Rubrics (Human) | - | 46.4 | 28.0 | 61.0 |
> > | &nbsp;&nbsp;&nbsp;&nbsp;+ OnlineRubrics-$\pi_\text{ref}$ | 8 (default) | 54.0 | 31.5 | 62.7 |
> > |                                                          | 4 | 53.0 | 29.1 | 62.0 |
> > |                                                          | 2 | 48.3 | 30.3 | 62.5 |
> > |                                                          | 1 | 48.2 | 28.7 | 62.3 |

---

### Official Review · Reviewer_sQcC · 2025-11-01

**Soundness:** 2
**Presentation:** 3
**Contribution:** 3
**Rating:** 6
**Confidence:** 4

**Summary:**

This paper addresses a critical limitation in rubric-based reinforcement learning for LLM post-training. The authors propose OnlineRubrics, a framework that dynamically elicits evaluation criteria during training by comparing responses from the current policy against a control policy. The key contributions are:
Novel Dynamic Rubric Elicitation Framework: A principled method for online criteria generation through pairwise response comparison, contrasting with static rubrics used in prior work (Gunjal et al., 2025; Viswanathan et al., 2025; Huang et al., 2025).
Theoretical Justification: Proposition 1 establishes that augmenting rubrics reduces the upper bound on gradient estimation error, formally motivating the approach.
Comprehensive Evaluation: Two curated datasets (Generalist and Expert Rubrics) with human-annotated, binary-verifiable criteria totaling 21,466 training rubrics and 10,941 evaluation rubrics.
Strong Empirical Results: Consistent improvements of 8-9% over static rubrics across multiple benchmarks (AlpacaEval win rate: 46.4%→55.0%, GPQA accuracy: 36.2%→38.1%, Arena-Hard: 52.4%→56.5%).
Qualitative Insights: Systematic analysis revealing that elicited criteria emphasize transparency, reproducibility, practicality, and anti-gaming properties.

**Strengths:**

1. Addresses a Real Problem with Practical Impact
The paper identifies a genuine limitation of static rubrics: vulnerability to reward hacking and inability to capture emergent behaviors. The "self-praising" example is particularly illustrative and motivates the work well. This problem becomes increasingly relevant as rubric-based training gains adoption.

2. Methodologically Sound Approach
The pairwise comparison strategy is well-motivated by preference learning literature (Bradley & Terry, 1952; Christiano et al., 2017). The intuition that discriminative comparison is easier than pointwise quality assessment is sensible and empirically validated (pointwise extraction baseline underperforms).

3. Rigorous Experimental Design
Two high-quality datasets with careful annotation principles (MECE, atomic, objective, self-contained)
Systematic verifier selection study (Figure 3) establishing GPT-4.1-mini as optimal cost-quality trade-off
Multiple strong baselines including synthetic rubrics, universal requirements, and pointwise extraction
Out-of-distribution evaluation on public benchmarks demonstrating generalization

4. Consistent and Substantial Improvements
Results are not cherry-picked—OnlineRubrics outperforms baselines across 8 of 9 evaluation metrics (Tables 2-3). The improvements on held-out evaluation sets (which contain no elicited rubrics) strongly suggest genuine capability enhancement rather than overfitting to generated criteria.

5. Transparency and Reproducibility
Excellent documentation in appendices: full system prompts (Figures 8-13), detailed experimental settings, data samples (Figures 5-6), and qualitative analysis with clustering. The two-stage extraction process (difference identification → criterion formation) is clearly described.

6. Valuable Qualitative Insights
The cluster analysis (Figure 7, Appendix E) reveals interpretable patterns: reproducibility (8.96%), practicality (8.33%), anti-gaming (7.69%). This provides actionable insights beyond numerical improvements.

**Weaknesses:**

1. Limited Model Diversity (Critical)
All experiments use a single base model (Qwen-2.5-7B-Instruct).
Impact: This severely limits claims about general applicability. The method might be model-specific or scale-specific.

2. Short Training Duration Raises Concerns
Only 3 epochs of training
Impact: Real-world deployment requires longer training. Without this analysis, practical utility is uncertain.

3. Computational Cost Not Adequately Addressed
Impact: Method may be impractical for resource-constrained settings or large-scale training.

4. Circular Dependency in LLM-Based Components
Both extractor (o3-mini) and grader (GPT-4.1-mini) are LLMs with known limitations.
Missing: Analysis of failure modes, error propagation, or human validation of elicited criteria quality.

5. Theoretical Contribution is Weak
Proposition 1 provides only an upper bound with limitations:
No analysis of bound tightness
Impact: Theory doesn't provide actionable insights or strong guarantees.

**Questions:**

Thank you to the authors for this well-executed work on an important problem—I look forward to seeing how this research develops with the suggested improvements.

1. Limited Model Diversity (Critical)
All experiments use a single base model (Qwen-2.5-7B-Instruct).
Key concerns with a specific case of a model:
Do results hold for other model families (Llama, Mistral, Gemma)?
Does the method work with smaller models (1-3B) where extractor quality may degrade?

2. Short Training Duration Raises Concerns
Only 3 epochs of training. Critical questions:
Does rubric quality degrade with longer training (10+ epochs)?
Is there rubric drift or accumulation of noise over extended training?
Do elicited criteria remain stable or do they become contradictory?
What happens to the total number of criteria—does it grow unboundedly?

3. Computational Cost Not Adequately Addressed
The method requires:
8 additional rollouts per prompt from control policy
8 LLM extractor calls (o3-mini) per prompt
1 deduplication LLM call per prompt
Additional grader calls for the expanded rubric set

Missing Analysis:
Total inference cost comparison with baselines (in dollars or FLOPs)
Wall-clock time increase
Sensitivity to number of pairwise comparisons (why 8?)
Cost-benefit trade-off analysis

4. Circular Dependency in LLM-Based Components
Both extractor (o3-mini) and grader (GPT-4.1-mini) are LLMs with known limitations. Concerns:
Do shared biases between extractor and grader create echo chambers?
Can the system amplify systematic errors over training iterations?
How does extractor quality affect final performance (no ablation study)?

---

> ### Author Response · Authors · 2025-11-24
> **Official Comment by Authors**
>
> Thank you for your detailed and thoughtful review. We are glad that you found the motivation and design of OnlineRubrics compelling, and the strength of our empirical and qualitative analyses convincing. Please find our responses to your concerns below.

---

> > ### Author Response · Authors · 2025-11-24
> > **Limited Model Diversity (W1 + Q1)**
> >
> > ### Base Model Diversity
> > We ran additional experiments using ``Llama-3.2-3B-Instruct`` with generalist and ``Llama-3.1-8B-Instruct`` with expert domains. The results, included in the new results tables above, show that OnlineRubrics continues to provide significant improvements over baselines across different model families and sizes. This suggests that our method is not model-specific and can generalize well to other LLMs.
> >
> > Please note that due to time and resource constraints, we have only tested OnlineRubrics-$\pi_\text{ref}$ on these additional models, which as discussed in the paper is a strong variant of our method.
> > | Model | Alpaca-Eval WR | Alpaca-Eval LC-WR | Generalist Rubrics |
> > |-------|-----------------|--------------------|------------------|
> > | Llama-3.2-3B-Instruct | 13.5 | 12.6 | 45.9 |
> > | &nbsp;&nbsp;+ LLM-Judge Score | 18.2 | 14.4 | 50.5 |
> > | &nbsp;&nbsp;+ Offline Rubrics (Human) | 19.4 | 14.4 | 51.6 |
> > | &nbsp;&nbsp;&nbsp;&nbsp;+ OnlineRubrics-$\pi_\text{old}$ | 20.6 | 16.2 | 52.0 |
> >
> > | Model | GPQA-D Acc. | GSM8K Acc. | Expert Rubrics |
> > |-------|-------------|------------|--------------|
> > | Llama-3.1-8B-Instruct | 27.2 | 56.7 | 25.1 |
> > | &nbsp;&nbsp;+ LLM-Judge Score | 31.7 | 62.4 | 34.3 |
> > | &nbsp;&nbsp;+ Offline Rubrics (Human) | 28.7 | 59.3 | 34.9 |
> > | &nbsp;&nbsp;&nbsp;&nbsp;+ OnlineRubrics-$\pi_\text{old}$ | 30.0 | 60.6 | 36.5 |
> >
> > ### Extractor Model Variations
> > We have experimented with various extractor models, including stronger models like `gpt-4.1` and `gpt-4.1-mini`, as well as an open-source model `gpt-oss-120B`. The new results tables above summarize the performance of OnlineRubrics with these different extractors. The results show that while stronger extractors (o3-mini, GPT-4.1) tend to yield better performance; smaller or open-source models still provide consistent improvements over optimizing against and LLM-Judge and `gpt-oss-120B` as an extractor results in competitive performance in Alpaca eval.
> >
> > | Model | Extractor | Alpaca-Eval WR | Alpaca-Eval LC-WR | Generalist Score |
> > | -------|-----------|-----------------|--------------------|------------------|
> > | Qwen-2.5-7B-Instruct | - | 30.0 | 28.2 | 55.4 |
> > | &nbsp;&nbsp;+ LLM-Judge Score | - | 42.2 | 26.9 | 58.5 |
> > | &nbsp;&nbsp;+ Offline Rubrics (Human) | - | 46.4 | 28.0 | 61.0 |
> > | &nbsp;&nbsp;&nbsp;&nbsp;+ OnlineRubrics-$\pi_\text{ref}$ | o3-mini | 54.0 | 31.5 | 62.7 |
> > |                                                          | gpt-4.1 | 49.9 | 25.4 | 61.9 |
> > |                                                          | gpt-4.1-mini | 48.9 | 28.0 | 61.8 |
> > |                                                          | gpt-4.1-nano | 50.5 | 27.7 | 60.4 |
> > |                                                          | gpt-oss-120B | 43.1 | 30.2 | 60.4 |

---

> > ### Author Response · Authors · 2025-11-24
> > **Short Training Duration (W2 + Q2)**
> >
> > We agree that long-horizon training is important for real-world deployment,
> > but we note that 3 epochs is already longer than the standard one or two epoch training used in many prior works [1, 2, 3, 4].
> > Importantly, the training curves in Figure 4 show that OnlineRubrics remains consistently above all baselines throughout all epochs.
> >
> > Also, we would like to clarify that extracted rubrics do not accumulate over epochs. At each step, OnlineRubrics extracts a fresh set of rubrics based on the current policy's behavior. Thus, there is no risk of unbounded growth in the number of criteria or accumulation of noise from previous epochs.
> >
> >
> > [1] Gunjal, A., Wang, A., Lau, E., Nath, V., He, Y., Liu, B., & Hendryx, S. (2025). Rubrics as Rewards: Reinforcement Learning Beyond Verifiable Domains. arXiv:2507.17746.
> >
> > [2] Viswanathan, V., Sun, Y., Ma, S., Kong, X., Cao, M., Neubig, G., & Wu, T. (2025). Checklists Are Better Than Reward Models for Aligning Language Models. arXiv:2507.18624.
> >
> > [3] Jacob Dineen, Aswin Rrv, Qin Liu, Zhikun Xu, Xiao Ye, Ming Shen, Zhaonan Li, Shijie Lu, Chitta Baral, Muhao Chen, and Ben Zhou. (2025). QA‐LIGN: Aligning LLMs through Constitutionally Decomposed QA. In Findings of the Association for Computational Linguistics: EMNLP 2025.
> >
> > [4] Allen Institute for AI. (2024). Olmo 3 Technical Report.

---

> > ### Author Response · Authors · 2025-11-24
> > **Computational Cost Overhead (W3 + Q3)**
> >
> > Thank you for raising this important point, we believe this is a valuable addition to our discussion.
> >
> > We would like to clarify that the computational cost overhead of OnlineRubrics primarily arises from LLM calls for rubric extraction and pairwise comparisons, not generation of the responses themselves. $\pi_\text{old}$ uses the rollouts generated by the policy for GRPO, and $\pi_\text{ref}$ uses rollouts from step 0 of GRPO, hence there is no additional cost in generating the responses.
> >
> > We acknowledge that there's a computational cost associated with: (1) rubric elicication and (2) grading with a longer rubric. Yet, this cost is within the same order of magnitude as verifying responses using LLM-based verifiers, which is a common practice in rubrics training [1, 2]. At each GRPO iteration, scoring responses against rubrics requires an LLM-judge to grade each response once. Let's denote the number of responses as N (we use N=16 rollouts in our experiments). Therefore, the cost of scoring responses is O(N).
> >
> > In OnlineRubrics, we use a pair of responses for each comparison, and we perform K comparisons per GRPO iteration (K=8 in our default setting). Thus, the cost of comparisons is O(2K). Also, note that by design $K \leq N$. Therefore, although there are 8+1 (8 comparisons + 1 final processing for deduplication) LLM calls per GRPO iteration in OnlineRubrics, the overall computational cost remains $O(N)$, which is within the same order of magnitude as traditional LLM-based verification methods.
> >
> > Note that the costs arising from (2) is negligible as the LLM grader evaluates all criteria in a single pass. The number of new extracted criteria are usually less than the existing criteria and as a whole, they are much shorter than the response itself, which is the majority of the computation time.
> >
> > In practice, a single baseline training costs $\approx$ 100 USD for rubric-based verification and $\approx$ 150 USD for OnlineRubrics with K=8 comparisons, indicating a 1.5x overhead. We believe this is a reasonable trade-off given the significant performance improvements demonstrated by OnlineRubrics.
> >
> >
> >
> > [1] Gunjal, A., Wang, A., Lau, E., Nath, V., He, Y., Liu, B., & Hendryx, S. (2025). Rubrics as Rewards: Reinforcement Learning Beyond Verifiable Domains. arXiv:2507.17746.
> >
> > [2] Viswanathan, V., Sun, Y., Ma, S., Kong, X., Cao, M., Neubig, G., & Wu, T. (2025). Checklists Are Better Than Reward Models for Aligning Language Models. arXiv:2507.18624.

---

> ### Author Response · Authors · 2025-11-24
> **Circular Dependency and Theoretical Contributions (W4 and W5)**
>
> ### Circular Dependency in LLM-Based Components (W4)
>
> We agree this is a valid point, though we observe consistent gains on GSM8K and GPQA-D, both with ground-truth answers, which suggests that the method remains robust even in the presence of potential LLM-generated errors.
>
> ### Theoretical Contributions (W5)
>
> Thank you for the thoughtful feedback. We clarify that this section is not intended to provide asymptotically tight RL optimization guarantees, rather it formalizes why eliciting missing criteria matters in rubric-based reward modeling. We note that because the dependency on missing criteria mass $||w_I||_1$ is unavoidable, missing latent criteria induces error in reward calculation which in turn induces gradient misalignment. We will revise the presentation of this section to make this intention more explicit.

---

### Author Response · Authors · 2025-12-04
**Summary of Discussions**

Dear AC,

Given the early closure of the discussion phase shortly after our responses were submitted, we would like to take this opportunity to summarize the key points from the reviews and our rebuttals for your consideration.

We’re pleased that reviewers found OnlineRubrics to be original (wkK2, sQcC, nonc) with strong empirical results (wkK2, sQcC, nonc, yBS8). It is encouraging that they found the motivation compelling (sQcC, wkK2, yBS8) and the paper to be well-written and clear (sQcC, nonc, yBS8).


We are grateful for the constructive feedback from all reviewers and have addressed their concerns in detail. Below is a summary of our responses to the common concerns raised:

* *Limited Model Diversity* (sQcC, yBS8):
We ran experiments with two additional base models (Llama-3.2-3B-Instruct and Llama-3.1-8B-Instruct). Results confirm that OnlineRubrics consistently improves over baselines across model families and sizes.

* *Extractor Model Variants* (sQcC, wkK2):
We ran experiments a range of different extractors. While stronger models yield better performance, even smaller or open-source extractors provide consistent improvements over baselines.

* *Computational Cost* (sQcC, wkK2):
We provided a breakdown of inference cost and showed that OnlineRubrics
remains within the same order of magnitude as offline rubric training with a reasonable trade-off for the significant performance gains.

We also addressed other reviewer-specific concerns, such as *the number of comparisons used for extraction* (nonc; ablation added), *theoretical justification limits* (sQcC; clarified motivation and scope of Proposition 1), and *rubric stability over training* (sQcC, nonc; addressed both qualitatively and quantitatively).

Please find our detailed responses to each reviewer below. We believe these additional experiments and clarifications strengthen our work and will add them to the camera-ready version of the paper.


Thank you for your time and consideration.

---

### Meta-Review · Area_Chair_JJ8D · 2026-01-07

**Summary:**

This paper tackles a key limitation of rubric-based reinforcement learning for LLM post-training: the reliance on static, pre-defined evaluation criteria. The authors introduce OnlineRubrics, a framework that dynamically elicits evaluation criteria during training by comparing responses from the current policy against a control policy.

Despite the novelty of the approach, there are several important concerns that are not adequately addressed:

1. Limited model diversity. While the authors include additional LLaMA variants, this setup does not meaningfully illuminate scaling behavior and offers only limited architectural diversity. Evaluating a broader range of model sizes and families would strengthen the empirical claims. Moreover, the paper does not compare against several existing reward modeling and RLHF baselines, many of which are directly relevant. Including such comparisons would better contextualize the benefits of OnlineRubrics.

2. Unclear source of performance gains. The origin of the observed improvements remains ambiguous. I agree with Reviewer nonc that further analysis is needed to determine whether the gains primarily arise from the dynamically elicited rubrics or from prompt-related effects. Additional ablation studies that isolate the contribution of rubric generation from prompt design would help clarify this issue.

I recommend the paper to not be accepted at this time and need to be revised for future submission.

Reference issue. The paper also contains a problematic reference that should not have occurred, but as requested by the PCs this considered as typograpical issue. Please don't repeat this mistake in the future.

Xueguang Li, Qian Liu, Dongfu Jiang, Ge Zhang, Zejun Ma, and Wenhu Chen. Gapo: Gradientadaptive policy optimization for multi-objective rlhf. arXiv preprint arXiv:2505.14652, 2025.

**Reviewer Concerns:**

1. Limited model diversity. While the authors include additional LLaMA variants, this setup does not meaningfully illuminate scaling behavior and offers only limited architectural diversity. Evaluating a broader range of model sizes and families would strengthen the empirical claims. Moreover, the paper does not compare against several existing reward modeling and RLHF baselines, many of which are directly relevant. Including such comparisons would better contextualize the benefits of OnlineRubrics.

2. Unclear source of performance gains. The origin of the observed improvements remains ambiguous. I agree with Reviewer nonc that further analysis is needed to determine whether the gains primarily arise from the dynamically elicited rubrics or from prompt-related effects. Additional ablation studies that isolate the contribution of rubric generation from prompt design would help clarify this issue.

**Reviewer Scores:**

N/A

---

### Decision · Program_Chairs · 2026-01-26

Reject